# Red light-driven electron sacrificial agents-free photoreduction of inert aryl halides via triplet-triplet annihilation

Le Zeng [1,2,3], Ling Huang[1,2,3], Wenhai Lin[1], Lin-Han Jiang[2] & Gang Han [1] ✉

Selective photoactivation of inert aryl halides is a fundamental challenge in organic synthesis. Specially, the long-wavelength red light is more desirable than the widely-applied blue light as the excitation source for photoredox catalysis, due to its superior penetration depth. However, the long-wavelength red light-driven photoactivation of inert aryl halides remains a challenge, mainly because of the low energy of the single long-wavelength red photon. Herein, we report the photoreduction of aryl bromides/chlorides with 656 nm LED via triplet-triplet annihilation (TTA) strategy. This method is based on our discovery that the commonly used chromophore of perylene can serve as an efficient and metal-free photocatalyst to enable the photoreduction of inert aryl halides without the conventional need for electronic sacrificial agents. By introducing a red light-absorbing photosensitizer to this perylene system, we accomplish the long-wavelength red light-driven photoreduction of aryl halides via sensitized TTA mechanism. Moreover, the performance of such a TTA-mediated photoreduction can be significantly enhanced when restricting the rotation freedom of phenyl moiety for perylene derivatives to suppress their triplet nonradiative transition, in both small and large-scale reaction settings.

Photoredox catalysis is fundamentally important in both the chemistry and materials field[1,2]. For example, the energy-demanding photo-activation of inert aryl halides to produce aromatic radical has attracted lots of attention in view of the prevalent existence and essential role of aromatic moiety in numerous fields, such as transition metal catalysis[3], asymmetric catalysis[4], the total synthesis of natural products[5], and polymer science[6,7]. However, the existing approach for the photoactivation of inert aryl halides requires high-energy photons such as ultraviolet (UV) and short-wavelength visible light (λ<500 nm)[8]. Due to the intense absorption of the involved substrates and inter-mediates in the short-wavelength area, the reaction penetration depth of high-energy short-wavelength light is quite shallow, leading to low product yield and inevitable byproducts, especially for large-scale reactions[9]. Certain engineering methods like micro-flow chemical

reactors were attempted to expand the interface area between visible light and photocatalyst[10]. Yet, such engineered micro channels also suffer from rather low overall reaction yield on a large-scale synthesis, especially for industrial-level production[11]. Moreover, the use of the micro-flow chemical reactor is tedious and complicated, requiring a finely-controlled flow rate of the reaction solution and reactants[11].

In contrast, the development of photoredox catalysis that can utilize the long-wavelength red light to drive a chemical reaction is highly desirable due to the superior penetration depth of long-wavelength red light and straightforward operation[9,12,13]. However, the energy of a single long wavelength photon is theoretically too low to initiate bond dissociation, especially for energy-demanding reactions, such as the photoreduction of inert aryl bromide/chloride[14]. For example, a blue photon with λ of 450 nm carries an energy of 2.8 eV,

[1]Department of Biochemistry and Molecular Biotechnology, University of Massachusetts Chan Medical School, Worcester, MA 01605, USA. [2]Research Center for Analytical Sciences and Tianjin Key Laboratory of Biosensing and Molecular Recognition, College of Chemistry, Nankai University, Tianjin 300192, China. [3]These authors contributed equally: Le Zeng, Ling Huang. ✉e-mail: Gang.Han@umassmed.edu

while a red photon with λ of 650 nm bears only 1.9 eV[12,15]. Recently, the stable radical anion-mediated multiphoton strategy was attempted for the photoactivation of inert aryl halides through the Z-scheme process[16–18]. In this process, electron-deficient photocatalysts were excited by light and then received an electron from sacrifice agents to form radical anions that absorb the second photon in order to attain enough energy[16–18]. Despite such progress, such Z-scheme style photoactivation of inert aryl halides have not been reported to be able to be conducted under long-wavelength red light[16–18]. Moreover, these methods require excessive electron sacrificial agents, which inevitably lead to a number of side reactions and complex purification[16–18]. The use of such environmentally toxic agents is also inconsistent with green chemistry requirements, especially for large-scale industrial-level productions[19,20].

Previously, short-wavelength triggered photocatalytic systems based on triplet-triplet annihilation (TTA) has been reported to reduce inert aryl halides with typically blue light at around 440 nm as the excitation source[8,16,21–24]. Such short-wavelength triggered TTA system also relies on sacrificial agents to operate the reduction[8,16,21–24]. Moreover, although near-infrared (NIR) TTA-UC systems were reported to be able to couple with visible light photocatalysts[9], the indirect energy transfer between the NIR TTA-UC pair and the visible light photocatalyst takes place via the rather inefficient emission-reabsorption process, which restricted their application for energy demanding photoreduction of inert aryl halides[9].

Herein, we report on a highly potent, electronic sacrificial agent-free, red light-driven photoreduction of inert aryl halides via a two-component TTA system, to address such a key bottleneck in visible-light-driven photoredox catalysis. The two components are the red

light absorbing-photosensitizer and the photocatalyst. In particular, this method is based on our discovery that the commonly used annihilator of perylene can serve as an efficient and metal-free photocatalyst to enable the photoreduction of inert aryl halides without the conventional need for electronic sacrificial agents via the oxidation quenching cycle (Supplementary Table 1). By introducing a long-wavelength red light absorbing photosensitizer, meso-tetraphenyl-tetrabenzoporphine Palladium complex (PdTPBP), we demonstrate that the energy of the two photons of red light can be directly transferred from PdTPBP to perylene in a highly efficient manner, generating the high-energy excited state of perylene via TTA process (Fig. 1a). In this way, long-wavelength red light-driven activation of inert aryl halides are achieved, which is not observed before. In our study, the upconverted singlet excited state of perylene via PdTPBP-sensitized TTA process directly participates in the catalytic cycle, rather than emits the upconversion emission (Fig. 1a). Moreover, the TTA-induced generation efficiency of singlet excited perylene can be improved via the systematical regulation of the steric hindrance of perylene derivatives (Py1–Py4) to suppress triplet nonradiative transition, which is evidenced by the significantly elevated TTA-UC efficiency up to 23% for PdTPBP/Py (Fig. 1b). As a result, satisfying photocatalytic results of energy-demanding activation of inert aryl halides were observed for this TTA mediated red light-driven system, which could facilitate the further reaction between generated aryl radicals with various coupling partners, even in large-scale volume (20 mL).

## Results

The oxidation potential of perylene was determined to be 0.90 V *vs.* SCE (Supplementary Table 2), suggesting the strong reducing ability of

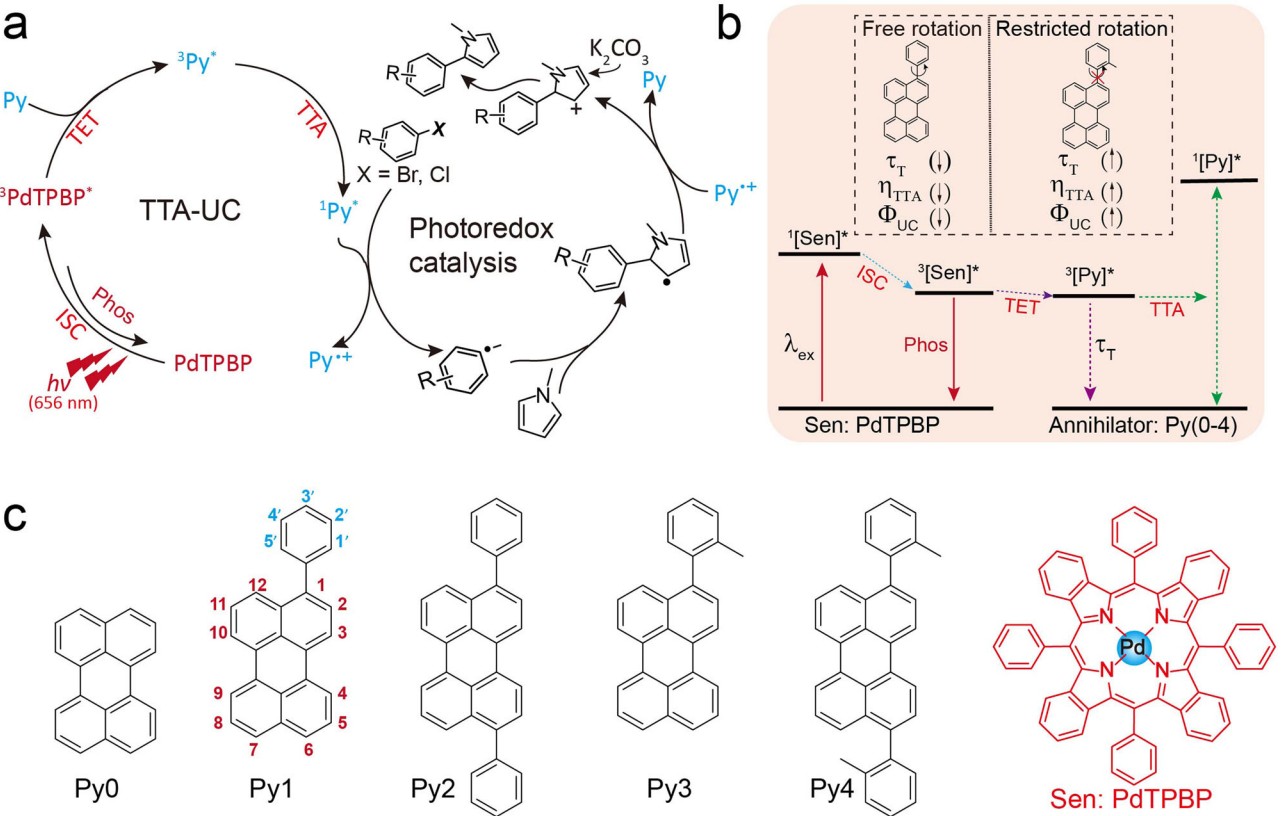

**Fig. 1 | Realization of red light-driven sacrificial agent-free photoactivation of aryl halides. a** The proposed mechanism of red light-driven sacrificial agent-free photoactivation of aryl halides via our TTA strategy; **b** Jablonski diagram showing the mechanism of restricting aryl rotation of Py annihilator to enhance TTA efficiency. **c** Molecular structures of photosensitizer PdTPBP and annihilators of Py0-

Py4. Sen sensitizer, Py perylene derivative, *hν* light irradiation, TET triplet energy transfer, Phos phosphorescence emission, ISC intersystem crossing, TTA triplet-triplet annihilation, $\tau_T$ low-lying triplet state lifetime, $\eta_{TTA}$ normalized TTA efficiency, $\Phi_{UC}$ upconversion quantum yield.

**Table 1 | Photoreduction of 4-bromoacetophenone and the cascade coupling with *N*-methylpyrrole in different conditions**

| Photocatalyst [a] | Light (nm) | Yield (%)[b] | Turnover number |
|---|---|---|---|
| Py0 | 455 | 69.5 | 34.8 |
| Py1 | 455 | 73.0 | 36.5 |
| Py2 | 455 | 59.6 | 29.8 |
| Py3 | 455 | 76.5 | 38.3 |
| Py4 | 455 | 70.3 | 35.1 |
| Py1 | 656 | Trace | – |
| PdTPBP | 656 | Trace | – |
| PdTPBP/Py0 | 656 | 36.9 | 18.5 |
| PdTPBP/Py1 | 656 | 53.7 | 26.8 |
| PdTPBP/Py2 | 656 | 59.5 | 29.8 |
| PdTPBP/Py3 | 656 | 68.0 | 34.0 |
| PdTPBP/Py4 | 656 | 79.4 | 39.7 |
| PdTPBP/Py4[c] | 656 | 67.0 | 37.5 |

Reaction condition: 4-bromoacetophenone (0.5 mmol), *N*-methylpyrrole (5.0 mmol), $K_2CO_3$ (1.0 mmol), DMSO (2 mL), blue or red light (60 mW/cm$^2$), in argon.

[a]Py (5 mM), PdTPBP (25 μM). For the structures of Py0–Py4 and PdTPBP please see Fig. 1.
[b]Isolated yield.
[c]The molar ratio of 4-bromoacetophenone and *N*-methylpyrrole is 1:1, both of which are 0.5 mmol.

excited perylene ($E_{red}$ = −1.88 V *vs* SCE), which enables perylene to be a potential photocatalyst for the photoreduction of inert aryl halides, such as 4-bromoacetophenone ($E_{red}$ = −1.84 V vs SCE)[25]. This hypothesis was verified by the successful photoreduction of 4-bromoacetophenone under blue light illumination with perylene, meanwhile the mechanism investigation showed that perylene was the photocatalyst and that no electron sacrificial agents are required (Supplementary Table 1). Control experiments in the absence of perylene, potassium carbonate, or light resulted in no conversion of 4-bromoacetophenone.

In view of the traditional role of perylene as the annihilator for the TTA-UC system, we selected the long-wavelength red absorbing photosensitizer PdTPBP (Supplementary Figure 1) for the sensitization of perylene, due to its intense red absorption and long triplet lifetime[26]. Accordingly, the subsequent TTA of sensitized perylene was expected to obtain highly reducing singlet excited state of perylene, leading to the activation of 4-bromoacetophenone (Fig. 1a). As shown in Table 1, under red light illumination of 656 nm with low power intensity at 60 mW/cm$^2$, no product was observed in the presence of only perylene or only PdTPBP, while a yield of 36.9% was detected when both were present, indicating that the above-mentioned hypothesis is feasible. However, compared to direct blue light excitation (69.5%), the performance of red light-driven TTA-mediated photoreduction of inert aryl halides was suboptimal.

To advance the photoreduction performance of our long-wavelength red light-driven TTA system, phenyl substituents were introduced to perylene for regulation of excited states (Fig. 1b). The steric hindrance of 2′-methyl was used to restrict the free rotation of the phenyl moiety (Fig. 1c). Both UV-Vis absorption and fluorescence spectra indicated the redshift of Py1-Py4 than Py0 (Supplementary Table 3). Owing to the presence of 2′-methyl, the absorption and fluorescence peaks in Py3 and Py4 were shorter than those of reference

molecules (Py1, Py2) (Fig. 2a). Additionally, Py1-Py4 showed high fluorescence quantum yield ($\Phi_f$ > 70%) and similar fluorescence lifetime to Py0, indicating that the aryl modification did not significantly affect the singlet excited states (Supplementary Table 3).

The electrochemical analysis of Py1-Py4 exhibited similar oxidation potential to perylene. In conjunction with the above-mentioned singlet state properties, this electrochemical result leads to similar values of excited-state reducing potential for perylene derivatives (Supplementary Table 2). As expected, Py1-Py4 exhibited good yields of the coupling products for 4-bromoacetophenone and *N*-methylpyrrole under blue light irradiation, suggesting that arylated perylene derivatives remain strong photoreduction ability (Table 1).

Next, theoretical calculation was used to gain insight into the photophysical properties of Py1-Py4. The energetically favorable geometric conformations of ground states were acquired via density functional theory (DFT) optimization. Steric hindrance of 2′-methyl results in the greater dihedral angle between phenyl moiety and perylene core for Py3 and Py4 than that of Py1 and Py2, respectively (Fig. 2b and Supplementary Figure 3). This steric hindrance-caused poor conjugation was further verified by the potential energy surfaces curve scanning (Supplementary Figures 4 and 5). As a result, compared with Py1 and Py2, the molecular frontier orbitals of Py3 and Py4 are mostly located at perylene core (Fig. 2c, Supplementary Figures 6 and 7).

Moreover, the calculated vertical excitation energies of Py1-Py4 are consistent with the experimental results of absorption spectra (Supplementary Table 4), confirming the reliability of the quantum mechanical modeling. Subsequently, the spin density calculation of the triplet state also suggested that the triplet state wave function of Py3 and Py4 are mainly located on the perylene core (Fig. 2d and Supplementary Figure 8). At the same time, the calculated $T_1$ state energy levels of Py3 and Py4 are higher than Py1 and Py2, respectively (Supplementary Table 4). These calculated results suggest that the steric hindrance of the substituents finely regulate the triplet and singlet excited states of perylene derivatives, which have the potential to improve the TTA-induced generation efficiency of singlet excited perylene derivatives and thus enhance their photoactivation efficiencies of aryl halides.

To investigate the steric hindrance influence on red light-driven photoactivation of aryl halides, the TTA properties of perylene derivatives (Py) with PdTPBP were explored first. Normally, for PdTPBP/Py in the absence of aryl halides, the generated high-energy singlet excited state of Py will return to its ground state via emission. Thus, the upconversion efficiency is an indicator of the generation efficiency of singlet excited state of Py, which directly determine the photocatalytic performance of our developed red light system of PdTPBP/Py. In this case, the TTA-UC performance of PdTPBP/Py was then assessed. In order to reduce its triplet quenching, the concentration of PdTPBP was set as 10 μM.

After optimization of the concentration for annihilator Py0-Py4 (Table 2 and Supplementary Figure 9), the power-dependent TTA-UC spectra were measured and analyzed to determine the excitation threshold density ($I_{th}$), where the upconverted emission intensity switches from a quadratic to a linear dependence on incident power. Above this threshold, the steady-state triplet population in the system is sufficiently high that TTA ceases to be the rate-limiting step[27]. The low threshold densities below 0.1 W/cm$^2$ were obtained for PdTPBP/perylene derivatives (Supplementary Figure 10), revealing the advantage of the utilization of low-power long-wavelength light for photoredox catalysis (Table 2). Moreover, the steric hindrances of substituents were found to improve the upconversion efficiency ($\eta_{UC}$), as Py3 and Py4 have higher $\eta_{UC}$ than their counterparts. In particular, the $\eta_{UC}$ of Py4 is as high as 23%, which is 3.2 folds higher than that of perylene (Table 2).

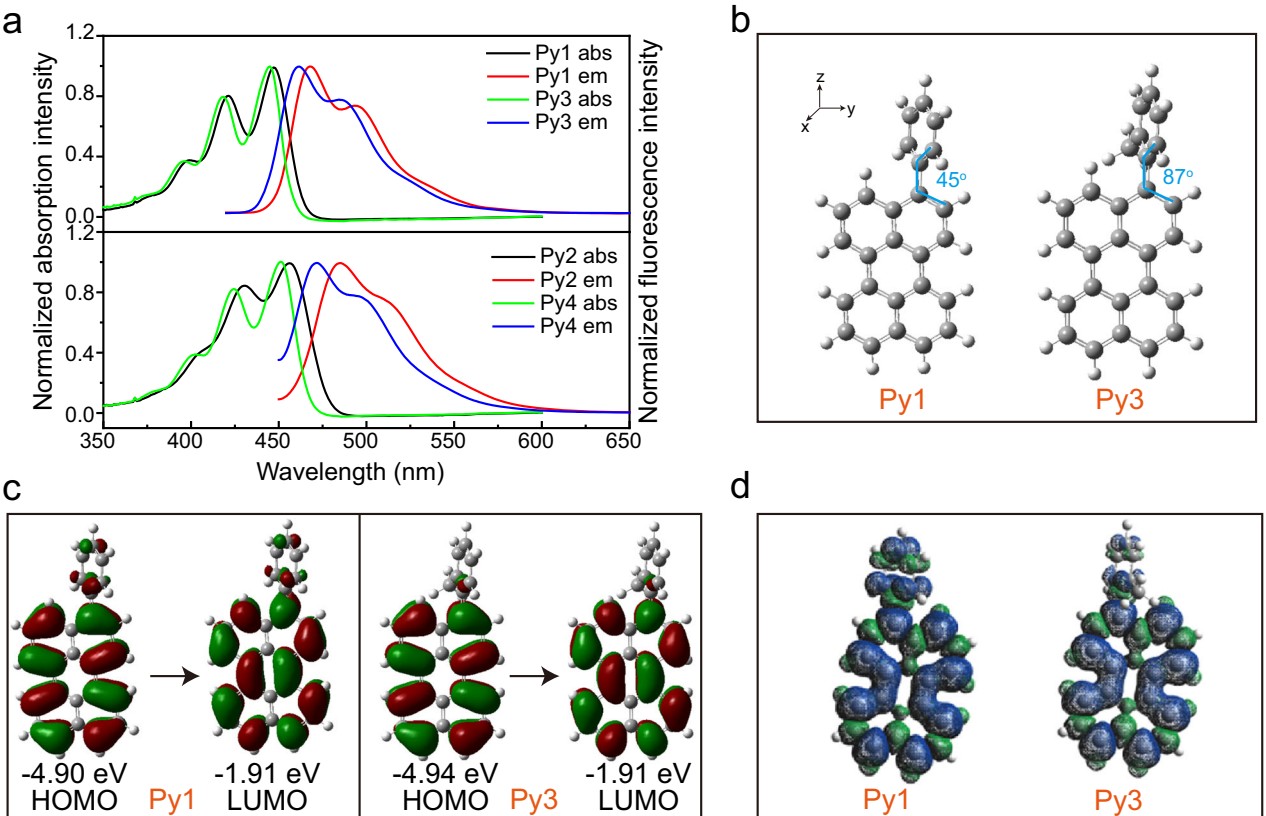

**Fig. 2 | Photophysical properties and theoretical calculation of perylene derivatives. a** Normalized UV-vis absorption (abs) and fluorescence emission (em) spectra of Py1-Py4 in toluene. **b** The optimized ground state conformations of Py1 and Py3 via DFT//B3LYP/6-31 G; **c** Electron density maps of the frontier molecular orbitals of Py1 and Py3; **d** Spin density surfaces of Py1 and Py3 in toluene at the optimized triplet state geometries.

**Table 2 | TTA-UC characteristics of annihilators (Py0-Py4) in combination with PdTPBP in toluene**

| An | $c$ [a] | $k_{sv}$ [b] | $k_q$ [c] | $I_{th}$ [d] | $\Phi_{TTET}$ [e] | $\eta_{TTA}$ [f] | $\Phi_{UC}$ [g] | $\eta_{UC}$ [h] | $\tau_{DF}$ [i] |
|---|---|---|---|---|---|---|---|---|---|
| Py0 | 700 | 11.0 | 7.8 | 81.4 | 97 | 4.3 | 3.5 | 7.0 | 374.1 |
| Py1 | 600 | 14.9 | 10.5 | 80.9 | 97.5 | 6.8 | 5.8 | 11.6 | 493.5 |
| Py2 | 500 | 24.0 | 16.9 | 80.4 | 98.3 | 14.0 | 9.8 | 19.6 | 539.9 |
| Py3 | 1000 | 12.7 | 8.9 | 97.6 | 98.1 | 11.3 | 9.3 | 18.6 | 607.3 |
| Py4 | 500 | 14.4 | 10.2 | 82.5 | 96.7 | 15.7 | 11.4 | 22.8 | 622.4 |

[a]Optimized concentration of annihilator (μM);
[b]Stern–Volmer quenching constant ($10^4 M^{-1}$);
[c]Bimolecular quenching constant ($10^8 M^{-1}s^{-1}$);
[d]Threshold power intensity (mW/cm²);
[e]Triplet-triplet energy transfer efficiency ($\Phi_{TTET}$ %);
[f]Normalized triplet-triplet annihilation efficiency ($\eta_{TTA}$ %);
[g] TTA-UC quantum yield (%);
[h] $\eta_{UC} = 2 \times \Phi_{UC}$, (%);
[i] Upconverted delayed fluorescence lifetime (μs).

The underlying reason for the superior upconversion efficiency of PdTPBP/Py4 was then further studied. The phosphorescence quenching of PdTPBP upon titration of perylene derivatives was measured to calculate the Stern–Volmer quenching constant ($k_{sv}$) and the bimolecular quenching constant ($k_q$) (Table 2). Arylated perylene derivatives (Py1-Py4) presented higher $k_{sv}$ values than Py0, which is likely due to the lower T$_1$ state after modification with phenyl moiety (Supplementary Figure 11). However, the TET efficiency ($\Phi_{TET}$) of all TTA-UC pairs are almost unity at the optimal concentration (Supplementary Figure 12), indicating that the triplet energy transfer step is not the key to the distinct $\eta_{UC}$ values (Table 2). On the other hand, the normalized TTA efficiency ($\eta_{TTA}$) of Py3 and Py4 is higher than Py1 and Py2,

suggesting that the TTA step is the determining factor for $\eta_{UC}$. In view of their analogous molecular weight and shape, the molecular diffusion of Py1-Py4 are similar in toluene, thus their $\eta_{TTA}$ depend on their triplet lifetime after sensitization[28]. As expected, Py3 and Py4 showed a longer delayed fluorescence lifetime than Py1 and Py2, respectively (Fig. 3e and Supplementary Figure 13). These results illustrate that the free rotation of phenyl moiety significantly quenched the lifetime of the triplet state. To further confirm this result, the temperature-dependent TTA-UC intensities were tested (Supplementary Figure 14). In general, the increasing temperature may be able to promote the intermolecular collision of photosensitizer/annihilator and annihilator/annihilator to enhance TTA-UC intensity[29].

However, an increase in temperature leads to fast free rotation rate of phenyl moiety (Py1 and Py2), intensifying the triplet nonradiative transition of the annihilator, then resulting in decreased upconversion efficiency. Thus, we compared the change of TTA-UC intensity with temperature increase of the same magnitude. As shown in Fig. 3f, the TTA-UC intensity of perylene derivatives with steric hindrance (Py3 and Py4) increased much more than that of Py1 and Py2, suggesting that the steric hindrance of 2′-methyl suppressed the free rotation-caused triplet states quenching, thus improving the $\eta_{UC}$.

Next, we test whether the increased $\eta_{UC}$ contributes to our red light-driven TTA-mediated photoactivation of aryl halides. Under previously optimized reaction conditions, the successful photoactivation of 4-bromoacetophenone with the combination of Py0-Py4 and PdTPBP, were observed under 656 nm LED illumination. PdTPBP/Py4 with highest $\eta_{UC}$ showed the highest product yield and turnover number of 79.4% and 39.7, whereas the original TTA-UC pair of PdTPBP/Py0 only gave a yield of 36.9%. In addition, PdTPBP/Py3 and PdTPBP/Py4 also presented a higher product yield and turnover

number than their counterpart, PdTPBP/Py1 (53.7%, 26.8) and PdTPBP/Py2 (59.5%, 29.8), respectively (Table 1). Moreover, the quantum yield for the overall photocatalytic reaction ($\Phi_{overall}$) is calculated with the established method[24], and the results are listed in Supplementary Table 5. To compare the $\Phi_{overall}$ for different TTA pairs (PdTPBP/Py0–PdTPBP/Py4), the concentrations of photosensitizer (PdTPBP), photocatalyst (Py0-Py4) and substrate 4-bromoacetophenone are fixed to be 10 μM, 500 μM and 50 mM respectively. The $\Phi_{overall}$ of PdTPBP/Py4 is 1.9%, which is higher than other TTA pairs. For each TTA pair, about 20% of the single excited state perylene derivatives was used for photoreduction of aryl halides. These findings confirm that restricting the rotation of the phenyl moiety can populate more singlet excited state of the photocatalyst/annihilator, thus improving the product yield. More interestingly, the red photoactivation of 4-bromoacetophenone with PdTPBP/Py4 (79.4%) displayed higher product yield than that of the direct blue photoactivation with Py4 (70.3%), even with the identical light intensity. This might results from the long-lived singlet excited state of Py4 via TTA mechanism, which owns a upconverted delayed fluorescence lifetime (510.3 μs) five orders of magnitude longer than the intrinsic fluorescence lifetime of Py4 (4.80 ns) in the argon-saturated DMSO, thus promoting the photo-induced electron transfer to substrate[30]. Additionally, photocatalysis with low-energy red light avoids the photobleaching of photocatalysts, especially compared to blue light exposure, which also contributes to improved photocatalytic efficiency.

The mechanism of TTA-mediated photoreduction of inert aryl halides was then investigated in depth. First, we calculated the Gibbs free energy of electron transfer ($\Delta G_{et}$) between the excited state of the photocatalyst and 4-bromoacetophenone by the Rehm-Weller equation using the oxidation potential of Py0-Py4, the reduction potential of 4-bromoacetophenone, and the singlet ($S_1$) and triplet ($T_1$) state energy levels of Py0-Py4 (Supplementary Table 2)[31]. It was found that thermodynamically supported electron transfer can only be achieved at the $S_1$ of the photocatalyst, which in turn generates Py radical cation. In addition, the ensuing reduction of Py4 radical cation back to neutral Py4 by the reaction intermediate is also thermodynamically supported with the negative $\Delta G$ value (see supporting information)[24]. Secondly, we have used the TTA pair PdTPBP/Py4 as an example to study the reaction kinetics of aryl halides photoreduction through the TTA mechanism. The Stern-Volmer constant ($k_{sv}$) of 5.0 $M^{-1}$ and the bimolecular constant ($k_q$) of 1.04 × $10^9 M^{-1} s^{-1}$ were recorded for 4-bromoacetophenone-induced TTA-UC quenching of PdTPBP/Py4[24]. However, 4-bromoacetophenone titration has no effect on the phosphorescence intensity and lifetime of PdTPBP (Supplementary Figures 15 and 16). Additionally, the luminescence lifetime of TTA-UC does not decrease in the presence of 4-bromoacetophenone, indicating that the triplet state of Py4 does not participate in the photoreduction of aryl halides (Supplementary Figure 15). Taken together, the TTA-UC quenching results from the quenching of singlet excited Py4 ($^1Py4^*$) by 4-bromoacetophenone, which is consistent with the thermodynamic analysis. The calculated activation barriers $\Delta G^{\neq}$ of 3.52 kJ/mol for the single electron transfer (ET) from $^1Py4^*$ to the 4-bromoacetophenone (see supporting information), also verified these results[24]. Meanwhile, to exclude the possibility of N-methylpyrrole serving as the electron sacrifice, we tried the photoreduction of 4-bromoacetophenone in the presence of only one equivalent N-methylpyrrole. Still, a 67% yield was detected for the coupling product (Table 1). This result clearly indicated that N-methylpyrrole is a reactant rather than the sacrificial agent, otherwise a yield of less than 50% is expected since the reaction with one molecule of 4-bromoacetophenone requires at least two molecules of N-methylpyrrole.

Using the optimized photocatalyst system of PdTPBP/Py4, the application range of our red light-driven sacrificial agent-free photoreduction of aryl halides was demonstrated via various coupling reactions between the generated aromatic radical and trapping reagents including pyrrole derivative, 1,3,5-trimethoxy benzene, indole, and triethyl phosphite. The product yields with blue-light photocatalyst Py4 were also listed for comparison (Fig. 4). The coupling between 4-bromoacetophenone and pyrrole derivatives, including N-methyl pyrrole, N-phenyl pyrrole, Boc-pyrrole and 2,4-dimethyl pyrrole, all had satisfying yields (> 60%) (compounds 1–4). Other trapping reagents also showed high photo-coupling yields toward 4-bromoacetophenone (compounds 5–7). In addition, aryl bromides with different substituents were evaluated, but high product yields were only observed for the strongly electron-deficient aryl bromides, such as bromoacetophenone, bromobenzaldehyde, and bromobenzonitrile (compounds 1, 8, 9). At the same time, both ortho and para-substituted aryl bromides can be well activated to give high product yields (compounds 9–10). A good product yield of 62.7% was also observed for electron-deficient heterocyclic bromide (compound 11). Moreover, the red light-driven Arbuzov reaction with electron-deficient aryl bromides were realized by using our TTA photoactivation system, and high yields of phosphonylation products were observed (compounds 12–15). Furthermore, the TTA-mediated photoactivation of aryl halides also enabled the facile late-stage modification of biologically active compounds without damage to other functional groups. As a result, the coupling between nicergoline and pyrrole or triethyl phosphite lead to the high product yield of 68.3% (compound 16) and 87.5% (compound 17) respectively. Notably, coupling yields of 14.0% and 17.2% are still achievable for triethyl phosphite with inert aryl chlorides, 2-chlorophenitrile and 4-chlorophenol, respectively (Fig. 4c).

For the above-mentioned coupling reactions, the red light driven photoactivation of aryl halides via a TTA mechanism showed comparable or higher yields than direct blue light photoredox catalysis. These results indicated that the synthesis approach of TTA-mediated photoactivation of inert aryl halides is convenient and efficient for preparing functional molecules. Finally, to demonstrate the penetration advantage of red light, the coupling reaction between 4-bromoacetophenone and triethyl phosphite was conducted in the reaction volume of 20 mL under the same power intensity and the same spot size of blue and red illumination. For blue light photocatalysis with only Py4, the product yield decreased from 83% to 39% when the reaction volume increased from 2 mL to 20 mL. In contrast, red light-driven photoactivation system with PdTPBP/Py4 did not show significant decrease in the product yield (66% of 20 mL vs 84% of 2 mL) (Fig. 4d). These distinct results illustrated that long-wavelength light is able to realize efficient large-scale photoredox catalysis via avoiding the side-absorption of incident light by substrates and reaction intermediates as well as the bleaching of the photocatalysts. These properties will further promote the applications of TTA-mediated photoredox catalysis to industrial production.

In conclusion, we reported a red light-induced electron sacrificial agents-free photoreduction of inert aryl halides via triplet-triplet annihilation. This is based on our unexpected finding that the perylene can serve as a highly potent and metal-free photocatalyst to induce the photoreduction of inert aryl halides without the need for electronic sacrificial agents. We then demonstrated that perylene enables the red light (656 nm) photoreduction of inert aryl halides without electronic sacrificial agents via the PdTPBP-sensitized TTA process. Moreover, restricting the rotation freedom of phenyl moiety for perylene derivatives to suppress their triplet nonradiative transition, can improve the photocoupling reaction between 4-bromoacetophenone and N-methyl pyrrole from 36.9% to 79.4%. The mechanism studies revealed that this photocatalytic improvement is originated from the efficiency increase of TTA-generated singlet excited state of perylene derivatives, as evidenced by the elevated TTA-UC efficiency of PdTPBP/Py from 7% to 23%. Furthermore, as red light can penetrate deep in the reaction solution, such photocatalytic reaction is highly efficient for large-scale volume

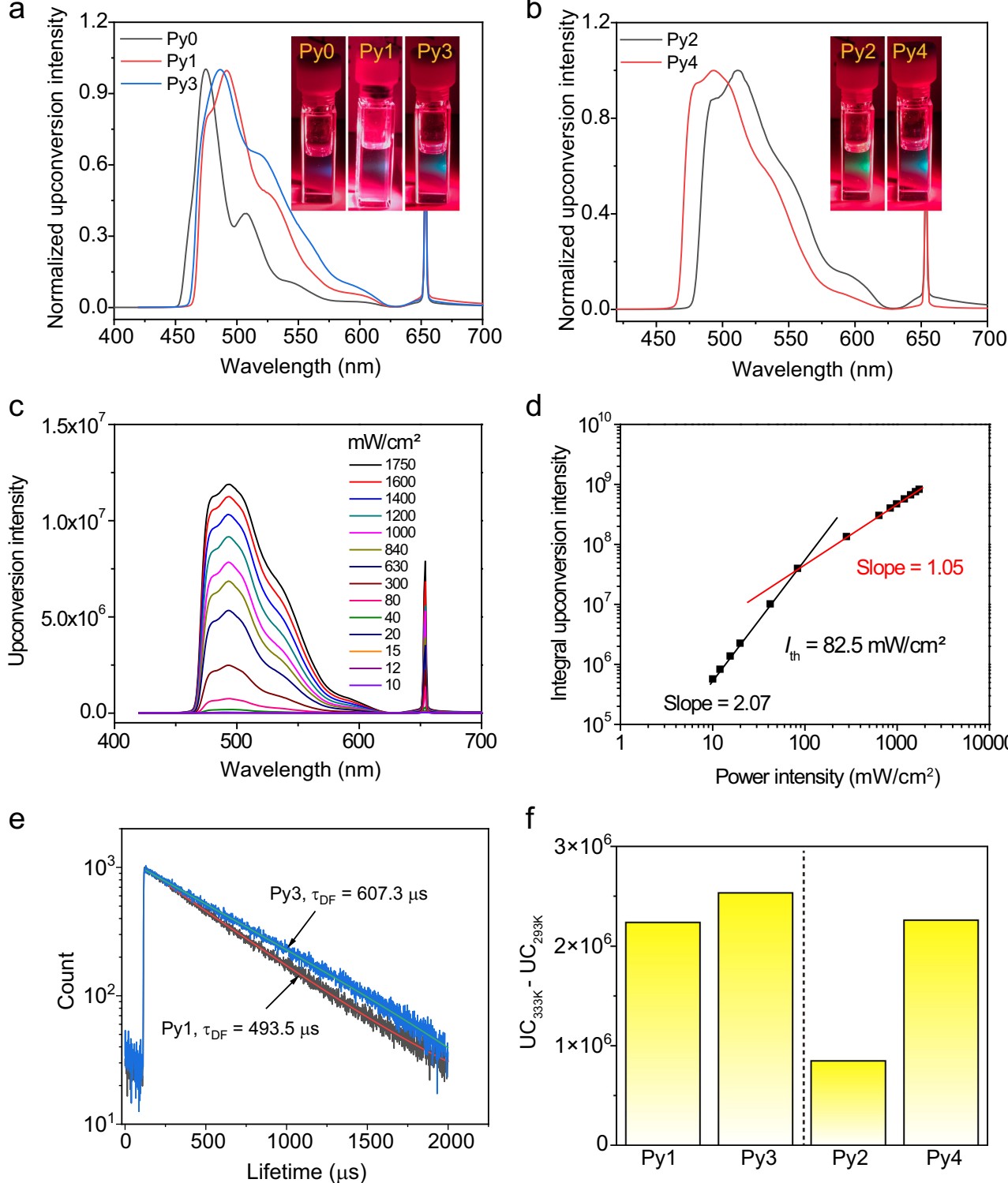

**Fig. 3 | Characterization of the TTA-UC properties for PdTPBP with Py0-Py4.**
Normalized TTA-UC spectra for PdTPBP with annihilators of (**a**) Py0, Py1, Py3 and
(**b**) Py2, Py4. The inserted images are the associated TTA-UC pictures.
**c** Upconversion emission spectra of PdTPBP/Py4 with various excitation intensities. **d** The dependence of the TTA-UC intensity of PdTPBP/Py4 on the incident
power density. The fitting lines have slopes of 2.07 (black, below) and 1.05 (red,

above) in the low- and high-power regions, respectively. The threshold power
intensity ($I_{th}$) is 82.5 mW/cm². **e** Upconverted delayed fluorescence lifetime ($\tau_{DF}$) of
PdTPBP/Py3 and PdTPBP/Py1 with a pulsed nanosecond laser at 653 nm as the
excitation light. **f** The change of upconversion (UC) intensity for Py1-Py4 in conjunction with PdTPBP from 293 K to 333 K. c (PdTPBP)= 10 μM, $\lambda_{ex}$ = 653 nm, in
argon saturated toluene.

reactions. This study not only overcomes the key roadblock in electron sacrificial agent-free photoredox catalysis, but also provides a
conceptual route to realize the industrial conversion of energy-demanding photocatalytic reactions.

## Methods
### Theoretical calculation
All theoretical calculations were performed using the Gaussian 09
program. The ground state geometries of the Py1-Py4 were optimized

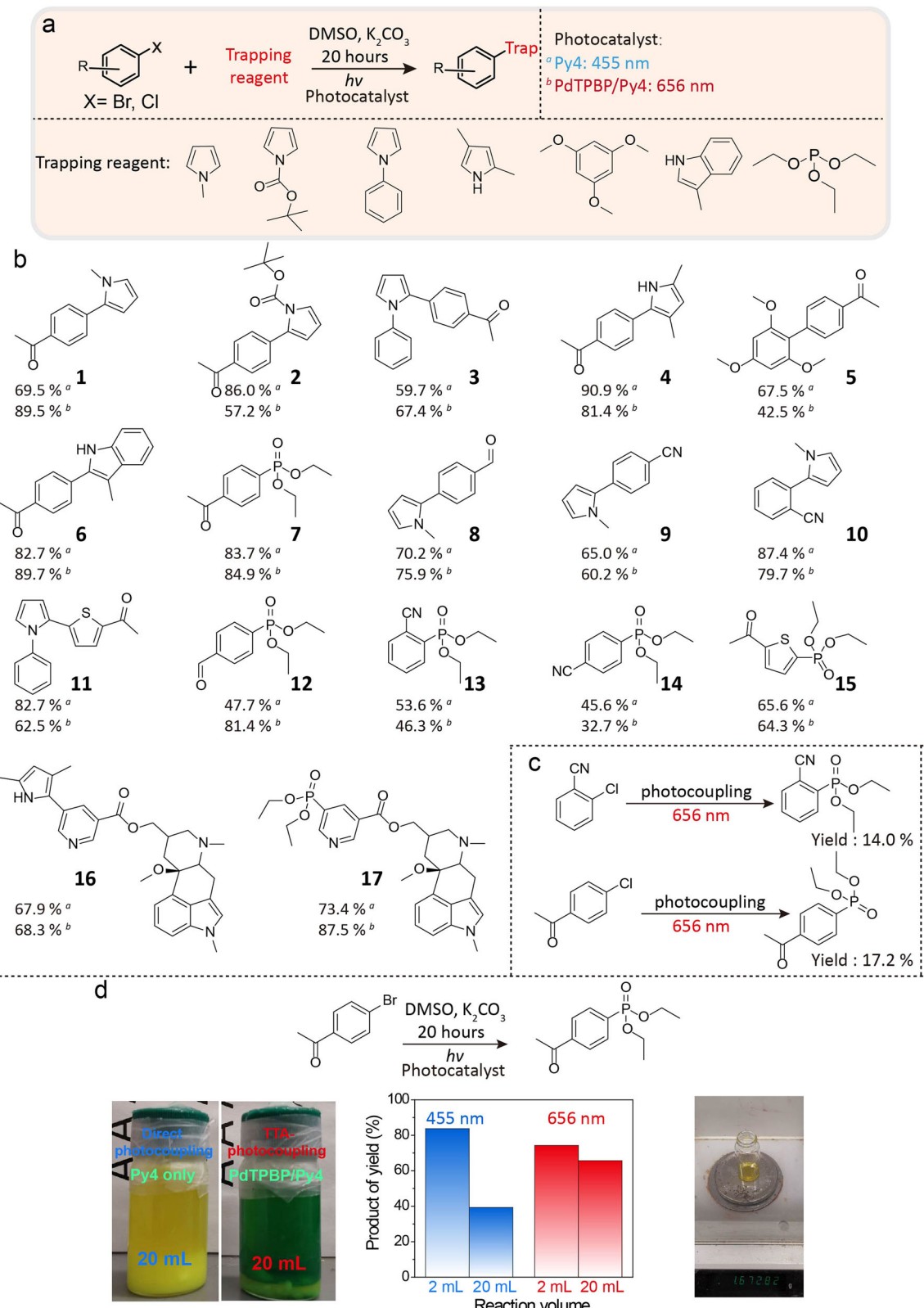

**Fig. 4 | Substrate scope of the red light-driven photoactivation of aryl halides via a TTA mechanism to give coupling products. a** The reaction formula of the investigated photocoupling between inert aryl halides with various trapping agents. **b** Photocoupling products for aryl bromides and the relative yields. **c** Photocoupling between aryl chlorides and triethyl phosphite. **d** The

photocoupling reaction of 4-bromoacetophenone and triethyl phosphite under different excitation wavelengths and reaction volumes. The left images are the 20 mL rection settings. The center figure is the product yield comparison. The right picture is the purified product after one 20 mL reaction driven by red light.

by density functional theory (DFT) based on B3LYP/6-31 G(d) level. Based on the optimized ground state geometry, the energies of the lowest singlet, triplet excited states, and isosurfaces of spin density were calculated by the TD-DFT method.

## General Procedure for the photoactivation of 4-bromoacetophenone with Py0-Py4 under blue light illumination (455 nm)

Photocatalyst (0.02 equiv.), inorganic base (3 equiv.), 4-bromoacetophenone (1 equiv.) and a magnetic stir bar were added to a dried 5 mL borosilicate vial. After purging the vial with argon, solvent (2 mL) and heteroarene (2 equiv.) were added by syringe. To complete the oxygen-remove for reaction mixture, three cycles of vacuum evacuation at −78 °C and then argon fill when returning to room temperature, was operated. Subsequently, the reaction vial was irradiated by a 455 nm LED placed approximately 2 cm far, with slow stirring (400 rpm). After a few hours, the reaction was quenched with water (10 mL) and extracted with DCM (×2), and then the combined organic layers were dried over MgSO$_4$. The crude material was then purified by column chromatography on silica gel using the mixture solvents of dichloromethane (DCM) and n-hexane (1/1, v/v) as elution solution to furnish the desired product.

## General Procedure for the long-wavelength light-driven photoactivation of aryl halides with Py0-Py4 via TTA strategy

Perylene derivatives Py (5 mM), PdTPBP (25 μM), K$_2$CO$_3$ (3 equiv.), inert aryl bromide/chloride (1 equiv.) and a magnetic stir bar were added to a dried 5 mL borosilicate vial. After purging the vial with argon, solvent (2 mL) and heteroarene (2 equiv.) were added by syringe. To complete the oxygen-remove for reaction mixture, three cycles of vacuum evacuation at −78 °C and then argon fill when returning to room temperature, was operated on the sealed vial. Subsequently, the reaction vial was irradiated by a 656 nm LED placed approximately 2 cm far, with slow stirring (400 rpm). After a few hours, the reaction was quenched with water (10 mL) and extracted with DCM (×2), and then the combined organic layers were dried over MgSO$_4$. The crude material was then purified by column chromatography on silica gel using the mixture solvents of dichloromethane (DCM) and n-hexane (1/1, v/v) as elution solution to furnish the desired product.

## Data availability

Supplementary Data 1: contains all the coordinates from the theoretical calculations.Other data supporting the findings of this study are available within the paper and its Supplementary Information files. Raw data can be obtained from the authors upon request.

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

## Acknowledgements

G. Han thanks for financial support provided by University of Massachusetts Chan Medical School.

## Author contributions

Conceptualization, G.H., L.H. and L.Z.; Methodology, L.H., L.Z., W.L. and L.J.; Investigation, L.H., L.Z., W.L. and L.J.; Visualization, L.Z. and L.H.; Supervision, G.H.; Writing original draft, L.Z. and L.H.; Writing review & editing, L.Z. L.H. and G.H.; Funding acquisition, G.H.

## Competing interests

The authors declare no competing interests.
