## [Peer Review File · Nature Communications]

Far-Red Light-Driven Electron Sacrificial Agents-Free Photoreduction of Inert Aryl Halides via Triplet-Triplet AnnihilationREVIEWER COMMENTS

Reviewer #1 (Remarks to the Author):

The authors report the activation of aryl halides by 650 nm light using a photocatalytic system consisting of a palladium sensitizer and a phenyl perylene as triplet-triplet annihilator. The efficiency of the reported system is good with synthetically useful yields and a reasonable large quantum yield.

The photophysical investigation of the reaction provides many details.

Still, I do not see the manuscript suitable for this journal for several reasons.

The synthetic chemistry is not new; several variants of the transformations have been described including efficient photochemical reactions.

The reaction is rather specialized and a broader application is not easily envisaged.

The catalytic system consist of two rather specialized components that have to be used in rather high concentrations (5 mM and 25 microM).

Overall, this is an interesting photophysical study and extension of previous reactions that should be of interest to a specialized group of readers. Publication in a journal focusing on photochemistry is recommended.

Reviewer #2 (Remarks to the Author):

The manuscript "Far-Red Light-Driven Electron Sacrificial Agents-Free Photoreduction of Inert Aryl Halides via Triplet-Triplet Annihilation" by gang han and co-workers describes the use of TTA to drive excited state activation of aryl halides using far-red light. This is an exciting piece of work since, in contrast to most previous works on TTA-UC that relies on upconverted photon emission and re-absorption, the current paper directly utilize the excited state to drive chemical reaction.

The manuscript is well structured and well written. The authors have engineered the perylene species with sterical groups to improve and better understand the process.

The authors have complemented the experimental work with DFT modelling, and also, a complete investigation of the TTA-UC properties of the system.

The scope of the reaction was tested on a large set of molecules.

All together, I think this a great contribution to the field. The manuscript is really well written, and all together, I recommend publication in the current form.

Reviewer #3 (Remarks to the Author):

These authors developed the long-wavelength far-red light driven photoredox catalysis for the selective activation of inert aryl halides via triplet-triplet annihilation (TTA) strategy. Universality and substrate specificity were well demonstrated, giving us an insight of the reaction based on the restriction of the rotation freedom of phenyl moiety for perylene derivatives to suppress the triplet nonradiative transition. However, some essential information as to photocatalysis is missing and these experiments need to be added before publication in Nature Commun.

(1) Missing the quantum yield for overall photocatalytic reaction. It is important to determine what percentage of the resulting input photons contributed to the photocatalytic reaction. In other words, what percentage of singlet perylene generated by TTA contributed to activate the phenyl halide?

(2) What is the turnover number of the perylene catalyst? The reviewer thought that recyclability is also an important indicator.

(3) Although the strong reducing ability of excited perylene for the photoreduction of inert aryl halides

was mentioned, the thermodynamics and kinetics of $\text{Py}^{\cdot-}$ reduction by the radical reaction intermediate have rarely been discussed, despite their importance in achieving a sacrificial agent-free photocatalytic reaction cycle.

(4) How can the authors eliminate the possibility that the reaction substrate has been used as a sacrificial agent?

REVIEWER COMMENTS

Reviewer #1 (Remarks to the Author):

The authors report the activation of aryl halides by 650 nm light using a photocatalytic system consisting of a palladium sensitizer and a phenyl perylene as triplet-triplet annihilator. The efficiency of the reported system is good with synthetically useful yields and a reasonable large quantum yield. The photophysical investigation of the reaction provides many details. Still, I do not see the manuscript suitable for this journal for several reasons. The synthetic chemistry is not new; several variants of the transformations have been described including efficient photochemical reactions. The reaction is rather specialized and a broader application is not easily envisaged. The catalytic system consist of two rather specialized components that have to be used in rather high concentrations (5 mM and 25 microM). Overall, this is an interesting photophysical study and extension of previous reactions that shuld be of interest to a specialized group of readers. Publication in a journal focusing on photochemistry is recommended.

Reply: We apologize if there may have been some misunderstanding in reading our manuscript, leading to your being less enthusiastic about it as compared to the other Reviewers. To address your concern and in regard to the significance of its broad applicability, we will discuss this in the subsequent paragraph.

Firstly, TTA-UC materials have emerged as the next-generation of upconversion materials. (*Acc. Chem. Res.* **2022**, *55*, 2604; *Chem. Rev.* **2021**, *121*, 9165; *Chem. Soc. Rev.* **2020**, *49*, 6529), Such materials have a wide variety of applications, such as those in photoredox catalysis (*Nature*, 2019, 565, 343), volumetric 3D printing (*Nature* 2022, 604, 474), as well as in biological areas (*Nat. Commun.* 2021, 12, 1898). Moreover, most previous works on TTA-UC rely on upconverted photon emission and re-absorption, as commented by Reviewer #2.

Secondly, as such an energy-demanding reaction from inert aryl halides has a wide variety of vital applications in medicinal chemistry, optoelectronic materials and polymer science, the development of far-red light driven highly efficient electron sacrificial agents-free photoreduction of inert aryl halides addresses a fundamental challenge in the realm of organic chemistry (*Science*, 2014, 346, 725; *Science*, 2016, 352,1082; *Nat. Catal.*, 2020, 3, 872).

In our TTA-UC system, unlike the traditional upconverted photon emission and re-absorption, the annihilator acts uniquely as the photocatalyst, thus considerably improving the utilization efficiency of photons for photoredox catalysis. Importantly, our approach does not need sacrificial agents that are currently required in the existing approaches (Table R1). The above advances are thus believed to be major improvements for photoredox catalysis in general, as well as for other application fields, which go well beyond the photochemistry field.

Table R1. Comparison of our work and the state-of-the-art systems for the photoreduction of inert aryl halides.

Photocatalyst (concentration) ^a	Sacrificial agent (dosage) ^b	Excitation Source	Reference
PDI (10 mol%, 3.3 mM)	Et ₃ N (8 equiv)	455 nm	Ghosh, I. et al. Science 2014 , 346, 725-8
Mes-Acr-BF ₄ (10 mol%, 30 mM)	DIPEA (3 equiv)	390 nm	MacKenzie, I. A. et al. Nature 2020 , 580, 76-80
3CzEPAIPN (5 mol%, 25 mM)	DIPEA (2 equiv)	456 nm	Xu, J. et al. J. Am. Chem. Soc. 2021 , 143, 13266-13273.
BD/PPO (130 mol%, 13 mM)	DMF (nd.)	430 nm	Majek, M. et al. Chem. Eur. J. 2015 , 21, 15496-501.
PtOEP/DPA (67 mol%, 6.7 mM)	DMF (nd.)	532 nm	Haring, M. et al. Chem. Commun. 2015 , 51, 16848-51.
[Cu(dap) ₂]Cl/DCA (10 mol%, 2.5 mM)	DIPEA (20 equiv)	623 nm	Glaser, F. et al. JACS Au 2022 , 2, 1488-1503.
PdTPBP/Py (2 mol%, 5 mM)	None	650 nm	This work.

^a For photocatalyst pair, the concentration of the one interacting with substrate is listed. The molar ratio is in relative to the aryl halide. ^b The amount of aryl halide is set as 1 equiv.

More specifically, the photoreduction of inert aryl halides is an energy-demanding and important reaction to afford aromatic radicals (Chem. Soc. Rev. 2021, 50, 2244; Nature 2020, 580, 76; Science 2014, 346, 725). As shown in Table R1, various efforts have been attempted to optimize this catalytic model by extending the excitation wavelength and reducing the dosage of the sacrificial agent. Our system stands out from others due to its having the longest absorption, and being completely sacrificial agent-free. Moreover, the photocatalyst-to-substrate mole ratio of PdTPBP/Py, 2 mol%, is the lowest among the photoreduction systems in related existing approaches.

Therefore, we hope that the Reviewer can now concur with our point that our method does indeed have notable benefits and would be of interest to both researchers and readers from a broad background, including organic synthesis, photochemistry, biology, and optoelectronic materials.

Reviewer #2 (Remarks to the Author):

The manuscript “Far-Red Light-Driven Electron Sacrificial Agents-Free Photoreduction of Inert Aryl Halides via Triplet-Triplet Annihilation” by gang han and co-workers describes the

use of TTA to drive excited state activation of aryl halides using far-red light. This is an exciting piece of work since, in contrast to most previous works on TTA-UC that relies on unconverted photon emission and re-absorption, the current paper directly utilize the excited state to drive chemical reaction. The manuscript is well structured and well written. The authors have engineered the perylene species with sterical groups to improve and better understand the process. The authors have complemented the experimental work with DFT modelling, and also, a complete investigation of the TTA-UC properties of the system. The scope of the reaction was tested on a large set of molecules. All together, I think this a great contribution to the field. The manuscript is really well written, and all together, I recommend publication in the current form.

Reply: We thank the Reviewer for this compliment with respect to our work. In addition, we are grateful that you recognized our work as “a great contribution to the field”.

Reviewer #3 (Remarks to the Author):

These authors developed the long-wavelength far-red light driven photoredox catalysis for the selective activation of inert aryl halides via triplet-triplet annihilation (TTA) strategy. Universality and substrate specificity were well demonstrated, giving us an insight of the reaction based on the restriction of the rotation freedom of phenyl moiety for perylene derivatives to suppress the triplet nonradiative transition. However, some essential information as to photocatalysis is missing and these experiments need to be added before publication in Nature Commun.

Reply: We thank the Reviewer for such insightful comments, which we will respond to one-by-one below.

(1) Missing the quantum yield for overall photocatalytic reaction. It is important to determine what percentage of the resulting input photons contributed to the photocatalytic reaction. In other words, what percentage of singlet perylene generated by TTA contributed to activate the phenyl halide?

Reply: With respect to your comment, the quantum yield for the overall photocatalytic reaction (Φ_{overall}) is calculated according to an established method (Carmen G. López-Calixto et al., Applied Catalysis B: Environmental, 2018, 237, 18–23), and the results are listed in Table R2/ Table S5. To compare the Φ_{overall} for different TTA pairs (PdTPBP/Py0–PdTPBP/Py4) in the photoreduction of inert aryl halides, the photosensitizer (PdTPBP) and photocatalyst (Py0–Py4) concentrations are fixed as 10 μM and 500 μM , respectively, and the 4-bromoacetophenone (50 mM), as the reaction substrate, was used to calculate the Φ_{overall} for each pair. Our results showed that, for each TTA pair, $\sim 20\%$ of the singlet excited state perylene derivatives was used for the photoreduction of aryl halides.

Table R2. The quantum yield for the overall photocatalytic reaction with PdTPBP/Py (Py0-Py4) as the photocatalyst pairs in DMSO.

An	Φ_{Py^*} (%) ^a	$\Phi_{\text{UC (without)}}$ (%) ^b	$\Phi_{\text{UC (within)}}$ (%) ^c	Φ_{overall} (%) ^d	contribution (%) ^e
Py0	3.5	2.9	2.3	0.6	20.7
Py1	5.3	4.6	3.7	0.9	19.6
Py2	11.0	7.9	6.3	1.6	20.2
Py3	8.9	7.4	5.9	1.5	20.2
Py4	12.8	9.7	7.8	1.9	19.6

^a Quantum efficiency of the generation of singlet excited Py (Py*) via TTA mechanism; ^b TTA-UC quantum yield in the absence of 4-bromoacetophenone. PdTPBP: 10 μM , Py: 500 μM . ^c TTA-UC quantum yield in the presence of 4-bromoacetophenone (50 mM). PdTPBP: 10 μM , Py: 500 μM . ^d The quantum yield for the overall photocatalytic reaction of Py0-Py4 in conjugation with PdTPBP. ^e The percentage of TTA-generated singlet excited Py for the activation of inert aryl halide.

(2) What is the turnover number of the perylene catalyst? The reviewer thought that recyclability is also an important indicator.

Reply: We agree with the Reviewer that the recyclability of photocatalysts is an important indicator.

According to the definition of turnover number:

$$\text{Turnover number} = \frac{\text{Moles of desired product formed}}{\text{moles of catalyst}}$$

The turnover numbers of the perylene catalyst (Py0-Py4) and TTA pairs (PdTPBP/Py0–PdTPBP/Py4) are now shown in **Table R3/****Table 1.**

Table R3. Photocatalytic performances of Py and their TTA-UC pairs for the photoreductive coupling between 4-bromoacetophenone and *N*-methylpyrrole.

Photocatalyst ^a	Light (nm)	Yield (%) ^b	Turnover number
Py0	455	69.5	34.8
Py1	455	73.0	36.5
Py2	455	59.6	29.8
Py3	455	76.5	38.3
Py4	455	70.3	35.1
Py1	650	Trace	–
PdTPBP	650	Trace	–
PdTPBP/Py0	650	36.9	18.5
PdTPBP/Py1	650	53.7	26.8
PdTPBP/Py2	650	59.5	29.8
PdTPBP/Py3	650	68.0	34.0
PdTPBP/Py4	650	79.4	39.7

Reaction conditions: 4-bromoacetophenone (0.5 mmol), *N*-methylpyrrole (5.0 mmol), K₂CO₃ (1.0 mmol), DMSO (2 mL), blue or red light (60 mW/cm²), in argon. ^a Py (5 mM), PdTPBP (25 μM). ^b Isolated yield.

(3) Although the strong reducing ability of excited perylene for the photoreduction of inert aryl halides was mentioned, the thermodynamics and kinetics of Py^{•+} reduction by the radical reaction intermediate have rarely been discussed, despite their importance in achieving a sacrificial agent-free photocatalytic reaction cycle.

Reply: We would like to thank the Reviewer for such constructive comments. According to these comments, the thermodynamics and kinetics of this reaction are now further discussed.

Thermodynamic analysis: First, we calculated the Gibbs free energy of electron transfer (ΔG_{et}) between the excited state of the photocatalyst and 4-bromoacetophenone by using the oxidation potential of Py0-Py4, the reduction potential of 4-bromoacetophenone, and the singlet (S_1) and triplet (T_1) state energy levels of Py0-Py4 based on the Rehm-Weller equation (Nancy Awwad et al, *Chem*, 2020, 6, 3071–3085). We found that thermodynamically supported electron transfer can only be achieved when the S_1 of the photocatalyst acts as an electron donor, which in turn generates Py radical cations ($\text{Py}^{+\bullet}$). The results are in **Table R4/**Table S2.

Table R4. Free energies for excited-state electron transfer from Py (S_1) and Py (T_1) to 4-bromoacetophenone.

Compound	E_{OX} (V) ^a	$E_{\text{OX}}(S_1)$ (V) ^b	$E_{\text{OX}}(T_1)$ (V) ^c	$\Delta G_{\text{et}}(S_1)$ (V) ^d	$\Delta G_{\text{et}}(T_1)$ (V) ^e
Py0	+0.90	-1.88	-0.63	-0.04	+1.21
Py1	+0.88	-1.85	-0.63	-0.01	+1.21
Py2	+0.63	-2.01	-0.85	-0.17	+0.99
Py3	+0.90	-1.85	-0.63	-0.01	+1.21
Py4	+0.81	-1.90	-0.71	-0.06	+1.13

^a Oxidation potential of Py0-Py4 vs. SCE. ^b Oxidation potential of Py0-Py4 in the singlet excited state (S_1). S_1 is estimated using the intersection of the absorption and emission spectra. ^c Oxidation potential of Py0-Py4 in the triplet excited state (T_1), T_1 energy level is estimated using TD-DFT calculation. ^d The Gibbs free energy of electron transfer between the photocatalyst and 4-bromoacetophenone occurs in the S_1 state of the photocatalyst, E_{red} (4-bromoacetophenone) = -1.84 V vs SCE. ^e The Gibbs free energy of electron transfer between the photocatalyst and 4-bromoacetophenone occurs in the T_1 state of the photocatalyst.

In addition, the back electron transfer process was analyzed according to the reported

protocol (Carmen G. López-Calixto et al., *Applied Catalysis B: Environmental*, **2018**, 237, 18–

23). We used photocatalyst Py4 as an example, and the calculation results are as follows:

Therefore: $\Delta G = -n \cdot F \cdot \Delta E$, $\Delta G < 0$ Exergonic process. This means the reaction intermediate can reduce the Py radical cation to the initial form to complete the catalytic cycle.

Kinetics analysis: we used the TTA pair PdTPBP/Py4 as an example to study the reaction kinetics of aryl halides photoreduction through the TTA mechanism. The Stern-Volmer quenching constant (k_{sv}) of 4-bromoacetophenone for the TTA-UC emission intensity of PdTPBP/Py4 was 5.0 M^{-1} ; the bimolecular quenching constant (k_q) is $1.04 \times 10^9 \text{ M}^{-1} \text{ s}^{-1}$, which presents the significant quenching for the delayed $^1\text{Py4}^*$ at a nearly diffusion-controlled rate. Additionally, the luminescence lifetime of TTA-UC does not decrease in the presence of 4-bromoacetophenone, indicating that the triplet state of Py4 does not participate in the photoreduction of aryl halides. This is consistent with the above thermodynamic analysis. Moreover, we used the established method (Carmen G. López-Calixto et al., *Applied Catalysis B: Environmental*, **2018**, 237, 18–23) to calculate the activation barrier (ΔG^\ddagger) for the singlet electron transfer from $^1\text{Py4}^*$ to 4-bromoacetophenone. The calculated ΔG^\ddagger is 3.52 kJ/mol, which is similar to that of other reported results.

Figure R1. Phosphorescent emission spectra of PdTPBP in the presence of 4-bromoacetophenone at different concentrations (0, 10, 20, 30, 40, 50 mM); (b) TTA-UC spectra of PdTPBP/Py4 at different concentrations of 4-bromoacetophenone (0, 10, 20, 30, 40, 50 mM); (c) Stern-Volmer quenching plotting of PdTPBP and PdTPBP/Py4 with 4-bromoacetophenone; (d) TTA-UC luminescence lifetime of PdTPBP/Py4 in the absence and presence of 4-bromoacetophenone. 4-bromoacetophenone, 50 mM. Solvent: argon-saturated DMSO, PdTPBP: 10 μM , Py4: 500 μM , $\lambda_{\text{ex}} = 650 \text{ nm}$.

Moreover, the addition of 4-bromoacetophenone has no effect on the phosphorescence intensity and lifetime of PdTPBP (Figure R2a or Fig. S16a). In contrast, under the irradiation of blue light, we observed the decrease of the fluorescence lifetime of Py4 from 4.93 to 4.76 ns in response to 4-bromoacetophenone (Figure R2b or Fig. S16a). This result indicated that $^1\text{Py4}^*$ can interact and reduce 4-bromoacetophenone to generate aryl radical. In sum, under far red light irradiation, only TTA-generated $^1\text{Py4}^*$ can reduce inert aryl halides.

Figure R2. (a) Phosphorescent lifetime of PdTPBP in the presence of different concentrations of 4-bromoacetophenone, $\lambda_{\text{ex}} = 650 \text{ nm}$. (b) The fluorescence lifetime of Py4 with or without 4-bromoacetophenone, $\lambda_{\text{ex}} = 405 \text{ nm}$. Solvent: argon-saturated DMSO, PdTPBP: 10 μM , Py4: 500 μM .

(4) How can the authors eliminate the possibility that the reaction substrate has been used as a sacrificial agent?

Reply: We thank the Reviewer for pointing this issue out. In our catalytic system, the only reaction substrate that might work as a sacrificial agent is *N*-methylpyrrole. In order to explore this possibility, we attempted to use 1-fold *N*-methylpyrrole as the reaction substrate. We observed a 67% product yield (**Table 1**, the last row, yellow highlight).

If *N*-methylpyrrole is an electron sacrifice, the reaction with one molecule of 4-bromoacetophenone requires at least two molecules of *N*-methylpyrrole. The yield of the reaction should be less than 50%. Since our observation (67%) is not consistent with such a hypothesis, the *N*-methylpyrrole is not an electron sacrifice in the reaction process.

REVIEWERS' COMMENTS

Reviewer #3 (Remarks to the Author):

After the careful revision and added experiments, this reviewer's comments were fully addressed. Current manuscript can be recommended to be published in Nature commun.